# Conservation of *Nematocida* microsporidia gene expression and host response in *Caenorhabditis* nematodes

**Yin Chen Wan**[1], **Emily R. Troemel**[2], **Aaron W. Reinke**[1]*

**1** Department of Molecular Genetics, University of Toronto, Toronto, Ontario, Canada, **2** School of Biological Sciences, University of California, San Diego, La Jolla, California, United States of America

* aaron.reinke@utoronto.ca

**Data Availability Statement:** Data availability All samples were deposited under NCBI BioProject PRJNA841614 and the sequence reads for all samples were submitted to the NCBI Sequence

## Abstract

Microsporidia are obligate intracellular parasites that are known to infect most types of animals. Many species of microsporidia can infect multiple related hosts, but it is not known if microsporidia express different genes depending upon which host species is infected or if the host response to infection is specific to each microsporidia species. To address these questions, we took advantage of two species of *Nematocida* microsporidia, *N. parisii* and *N. ausubeli*, that infect two species of *Caenorhabditis* nematodes, *C. elegans* and *C. briggsae*. We performed RNA-seq at several time points for each host infected with either microsporidia species. We observed that *Nematocida* transcription was largely independent of its host. We also observed that the host transcriptional response was similar when infected with either microsporidia species. Finally, we analyzed if the host response to microsporidia infection was conserved across host species. We observed that although many of the genes upregulated in response to infection are not direct orthologs, the same expanded gene families are upregulated in both *Caenorhabditis* hosts. Together our results describe the transcriptional interactions of *Nematocida* infection in *Caenorhabditis* hosts and demonstrate that these responses are evolutionarily conserved.

## Introduction

Microsporidia are obligate eukaryotic intracellular pathogens [1]. This fungal-related phylum contains over 1400 described species that infect a wide range of animal hosts including invertebrates, vertebrates, and protists [2, 3]. Although as a phylum microsporidia infect a wide range of hosts, most species only infect one or several closely related hosts [4, 5]. Throughout evolution, microsporidia have lost many metabolic and biosynthesis genes that are present in other eukaryotes [6]. These adaptations to survive within their hosts have resulted in microsporidia having the smallest known eukaryotic genomes [7]. These features of microsporidia being able to specifically infect most types of animals with a limited coding capacity have made microsporidia a powerful model for understanding the evolution of intracellular parasites [8–11].

Read Archive. All other relevant data are within the paper and its Supporting information files.

**Funding:** Canadian Institutes of Health Research grant no. 400784 (to A.R.) Alfred P. Sloan Research Fellowship FG2019-12040 (to A.R.) NIH R01's AG052622 and GM114139 to E.T. The funders had no role in study design, data collection and analysis, decision to publish, or preparation of the manuscript.

**Competing interests:** The authors have declared that no competing interests exist.

The model nematode *Caenorhabditis elegans* has become an important system for studying microsporidia infections [12]. *Nematocida parisii* and *Nematocida ausubeli* (referred to as *Nematocida* sp. 1 in some publications) are the two microsporidia species most commonly observed to infect *Caenorhabditis elegans* [13, 14]. These two species of *Nematocida* are also commonly found to infect *Caenorhabditis briggsae*, a related species which has been used as a model organism to facilitate comparative analyses to *C. elegans* [15, 16]. These two nematode species live in overlapping geographical locations [17, 18]. Both of these species are hermaphrodites, but they have some differences in development and response to temperature [19, 20]. *N. parisii* was first found infecting *C. elegans* outside of Paris and *N. ausubeli* was originally found infecting *C. briggsae* in India. Since then, both microsporidia species have been found infecting both hosts in Europe [13, 21].

The infection cycle of *N. parisii* and *N. ausubeli* in *Caenorhabditis* hosts begins with ingestion of the microsporidia spores, an environmentally resistant and dormant stage of the pathogen [22, 23]. In the intestine, the sporoplasm in the spore is ejected through the polar tube and deposited into the intestinal cells. This is followed by replication into meronts and differentiation into new spores, which then exit intestinal cells into the intestinal lumen. The spores are defecated by the nematodes, and the infection cycle repeats when they are ingested by another host [22, 24, 25].

There are some similarities in the infections caused by these two *Nematocida* species; they both exclusively infect the intestine, they infect *C. elegans* and *C. briggsae*, they cause intestinal cells to fuse, and they have similar life cycles [13, 14, 26]. Both species also use similar types of secreted and membrane bound proteins to interface with host proteins [21]. Although the divergence time between *N. parisii* and *N. ausubeli* is unknown, the two *Nematocida* species share 68.3% amino acid identity [24, 25]. These two species also have distinct growth and phenotypic characteristics during infection in *C. elegans*. For example, *N. ausubeli* displays faster growth and increased impairment of host fitness [26].

In response to microsporidia infection, hosts often display large transcriptional changes. The gene expression of *C. elegans* in response to *N. parisii* infection is reported to be distinct from responses to bacteria, but similar to nodavirus infection [27]. Although the response to *N. parisii* infection also shares some similarities to the response against fungi such as *Drechmeria coniospora* and *Harposporium*, the genes upregulated during *N. parisii* infection are still mostly specific to either microsporidia or viral infection [27]. The unique transcriptional response against *Nematocida* has been termed the intracellular pathogen response (IPR) [28]. Among upregulated IPR genes, many contain F-box, FTH, and MATH domains that are implicated in substrate recognition during ubiquitin-mediated degradation [27, 29]. The IPR also includes upregulation of several *Caenorhabditis* specific families that are still poorly understood, such as the PALS family [30]. Mutants that cause the IPR to be upregulated are resistant to infection and can clear *N. parisii* infections [31–33].

Studies looking at infection of nematodes by different bacterial and fungal pathogens showed both significant overlap between pathogen infections, as well as species-specific responses [34–36]. In addition, recognition of two species of oomycete displayed a shared transcriptional response [37]. Similarities of responses between different host species have also been observed. For example, the response to nodavirus infection is conserved between *C. elegans* and *C. briggsae* [38]. An intergenerational transcriptional response to *Pseudomonas vranovensis* was conserved between some, but not all species of *Caenorhabditis* [39]. A study looking at four different species of bacteria infecting the nematode *Pristionchus pacificus* also showed both similarities and differences between the transcriptional responses in the two hosts [40]. For example, while genes in the FOXO/DAF-16, TGF-beta and p38 MAP kinase pathways are suggested to be conserved between *C. elegans* and *P. pacificus* in response to

bacterial infection, host-specific transcriptional responses are also observed, such as the enrichment of lipid metabolism related domains in the response of *P. pacificus* to different bacteria [40]. Although conservation of responses between pathogen and hosts in *C. elegans* is observed, different strains of bacteria can elicit different responses, and different strains of *C. elegans* can have distinct responses [36, 41].

To determine the extent that microsporidia gene expression is influenced by its host and how conserved the host response is to microsporidia infection, we generated and analysed transcriptional data of *N. parisii* and *N. ausubeli* infecting *C. elegans* and *C. briggsae*. For the hosts, we used *C. elegans* and *C. briggsae*, which diverged ~18 million years ago [42]. About 60% of the genes from these species are orthologous and the median percent identity between these proteins is 80%, which is approximately the same extent of divergence between humans and mice [15]. We chose these two host models because both are infected by *N. parisii* and *N. ausubeli* efficiently in a laboratory setting, which recapitulates the natural interactions that exist in the wild [13, 14]. We observe similar transcriptional patterns of each *Nematocida* species in the two species of hosts with only a small set of differentially regulated genes. The host response to either of the two microsporidia species was also similar. This transcriptional response is conserved across *C. elegans* and *C. briggsae*. Altogether, our results suggest that different *Nematocida* species do not have distinct expression programs depending on the host, and transcriptional responses of *Caenorhabditis* hosts to *Nematocida* infection are conserved.

## Materials and methods

### Infection of nematodes with microsporidia

*C. elegans* strain N2 and *C. briggsae* strain AF16 were maintained at 21˚C on 10-cm nematode growth medium (NGM) plates seeded with *Escherichia coli* OP50-1 for at least three generations without starvation. Three plates each of mixed stage populations of animals were washed with M9 and embryos extracted by treating with sodium hypochlorite/1 M NaOH for 2.5 minutes. Embryos were hatched by incubating for 18 hours at 21˚C. ~30,000 L1s of each species were placed on a 10-cm plates along with 1 ml M9 and 10 μl of 10X OP50-1. Three plates of each species were prepared. Animals were infected with either 20 million *N. parisii* (ERTm1) or *N. ausubeli* (ERTm2) spores. These spores were prepared as described previously [22]. Uninfected plates were used as the control. Plates were dried for 1 hour in a clean cabinet and incubated an additional 1.5 hours at 21˚C. Animals were then removed from plates using two 5 ml washes of M9. After washing samples twice with 10 ml M9, all but 1 ml of M9 was removed from the samples. An additional 3 ml of M9 and 1 ml of 10X OP50-1 was added to each sample, and 1 ml of this solution was plated onto 4 10-cm plates per sample, resulting in ~6,000 animals per plate. At either 10, 20, or 28 hours post infection, plates were harvested by washing off worms 3 times with 725 μl M9. Samples were then washed 3 times with 1 ml M9/0.1% Tween-20.

To determine the extent of infection, ~1,000 animals from each sample were removed, fixed, and stained using an *N. parisii* 18S rRNA fluorescent in situ hybridization probe as previously described [21]. Vectashield containing DAPI (Vector Labs) was added and samples imaged using an Zeiss Axioimager 2.

### RNA extraction and sequencing

After the last wash, 1 ml of TRIzol was added to each sample and RNA was extracted similar to as described previously [27]. mRNA libraries were prepared by the UCSD IGM Genomics Center using Illumina Stranded mRNA Prep kit and sequenced on a single lane of an Illumina HiSeq 4000, using 100 bp paired-end reads.

## Microsporidia gene expression analysis

The respective *N. parisii* and *N. ausubeli* genome annotation files were converted into.gtf format using Gffread v0.12.3. Reads from each sample were mapped to either *Nematocida parisii* strain ERTm1 (Genbank: GCF_000250985.1) or *Nematocida ausubeli* strain ERTm2 (Genbank: GCA_000250695.1) using TopHat v2.1.2, which also calculates the overall read mapping rate [43]. Using FastQC, quality control was performed on the Tophat aligned files to ensure that they are of good quality (quality scores across all bases >30) and that the adapters were properly removed from the sequencing datasets prior to downstream analysis [44]. Transcriptome assemblies were then generated using Cufflinks v2.2.1 [45] and Cuffmerge in the Cufflinks package. Differentially expressed genes were determined using Cuffdiff v2.2.1 [46] and were visualised using the R package cummeRbund v2.36.0 [47].

The Fragments Per Kilobase of transcript per Million mapped reads (FPKM) values of *N. parisii* and *N. ausubeli* transcripts in each *Caenorhabditis* host were calculated by Cuffdiff v2.2.1 [46]. The linear correlation between gene expression in each host was determined using the R packages ggplot2 v3.3.5 and ggpubr v0.4.0; the ratio of FPKM values between the two hosts was calculated by dividing the microsporidia's FPKM values in *C. elegans* by that in *C. briggsae*. Hierarchical clustering of log10-transformed FPKM+1 values was done using the hclust function from the R package stats v4.1.1; heatmaps were generated by ggplot2 v3.3.5 in R.

## *Caenorhabditis* RNA-seq analysis

The paired end reads of each 10 hours post infection sample were submitted to Alaska v1.7.2 (http://alaska.caltech.edu), an online automatic *C.elegans* RNA-seq analysis pipeline. During the pipeline, briefly, Bowtie2 [48], Samtools [49], RSeQC [50], FastQC [44], and MultiQC [51] were used for quality control of the input files. Then, Kallisto [52] was used for read alignment and quantification, followed by differential analysis using Sleuth [53]. For *C. elegans* samples, the reads were aligned to the N2 reference of the WS268 release (accession: PRJNA13758); the reads from *C. briggsae* samples were aligned to reference genome of the WS268 release (accession: PRJNA10731). The 20 and 28 hours samples were not submitted for analysis due to the lack of replicates at these timepoints.

Log2 fold-change values of duplicated genes in each of the eight samples were averaged using the R package dplyr v1.0.8. Genes with FDR-adjusted p-value of <0.05 were regarded as significant. Differentially upregulated genes were defined as those with an FDR-adjusted p-value <0.05 and log2 fold change ≥0 (infected vs. control); differentially downregulated genes were defined as those with an FDR-adjusted p-value <0.05 and log2 fold change ≤0 (infected vs. control).

## Principal component analysis

The normalised abundance measurements in *C. elegans* and *C. briggsae* generated by Alaska, were read by the readRDS() function of the R package base v4.1.1. Normalized counts for each gene in each host species were generated via sleuth_to_matrix() with the "obs_norm" data and "tpm" units. PCA of gene expression was performed using samples from each species via the R package pcaexplorer v2.20.2 [54].

## Determination of gene expression overlap

Genes with an FDR<0.05 were used to compare expression overlap. Values of duplicated genes in each sample were averaged using the R package dplyr v1.0.8. Overlap between samples

was determined using the R package gplots v3.1.1. The p-values were calculated using the Fisher exact test. Statistical enrichment tests of shared and non-shared genes were performed using PANTHER on pantherdb.org [55]. Each input list was statistically tested for enrichment against the annotation sets: "GO biological process complete", "GO cellular component complete", "GO molecular function complete", "PANTHER GO-Slim Biological Process", "PANTHER GO-Slim Cellular Component", "PANTHER GO-Slim Molecular Function", "PANTHER Pathways", "Reactome pathways" and "PANTHER Protein Class". Results from these tests with p-value <0.05 are in S4 Table.

To determine overlap in our samples compared to differentially regulated genes from Bakowski *et al.*, we first used Alaska to process the read files of *C. elegans* infected by *N. parisii* samples at 8, 16, 30, 40 and 64 hours. To compare our samples' expression with data from Bakowski *et al.*, genes expressed in at least four of those samples were used for hierarchical clustering by hclust function in the R package stats v4.1.1. The dendrogram in the was produced using the R package ggdendrogram v0.1.23; the heatmap was generated by ggplot2 v3.3.5. Overlap between our samples, the Bakowski *et al.*, (2014) samples, and the Chen *et al.* (2017) samples containing differentially regulated genes for *C. elegans* N2 infected by Orsay virus or *N. parisii*, and *C. briggsae* infected by Santeuil virus were calculated as described above.

## Domain enrichment analysis

The type of gene classes and domains investigated for the enrichment analyses are listed in S6 Table. Among these, we actively chose to look at F-box, MATH/BATH, PALS, DUF684, DUF713, C-type lectins, and *skr* domains containing genes, implicated from differentially regulated genes of *N. parisii* infected *C. elegans* in previous publications [27, 38]. We also examined additional types of genes and domains enriched in our datasets using DAVID Bioinformatics Resources (2021) [56, 57] for *C. elegans* and the WormBase simple Ggne queries tool for *C. briggsae*. A list of gene names from respective gene classes were downloaded separately from Wormbase (http://wormbase.org/) and the corresponding protein-coding sequences were extracted from Wormbase ParaSite (https://parasite.wormbase.org/). Alignments of other domains or classes were downloaded directly from Pfam (http://pfam.xfam.org/) separately. After deleting duplicated sequences, protein sequences of these classes of genes were converted to Stockholm format using Clustal Omega Multiple Sequence Alignment tool (https://www.ebi.ac.uk/Tools/msa/clustalo/). Hmmbuild of HMMER v3.3.2 [58] was used to build respective profile HMMs, which was then searched against *C. elegans* (accession: PRJNA13758) proteome of WS268 release using hmmsearch. Output genes with E-value <1e-5 were used as the gene list for the enrichment analyses. In the output genes, MATH and BATH genes were combined into one list; *fbxa*, *fbxb* and *fbxc* genes were merged into the gene list for F-box while genes with CUB and CUB-like domains were categorised together (S6 Table). Using the R package gplots v3.1.1, the number of genes overlapping between the lists and samples were computed, then plotted using R package ggplot2 v3.3.5. Genes that do not fall into any of these gene classes or have any of these domains were categorized as "other".

## Determination of orthologs between *C. elegans* and *C. briggsae*

Orthologous genes were determined between *C. elegans* (accession: PRJNA13758) and *C. briggsae* (accession: PRJNA13758) proteomes of the WS268 release using Orthofinder v2.5.2 with the default settings [59].

### Single copy orthologs analysis

Genes that are single copy orthologs were extracted, based on the single copy orthologs list computed by Orthofinder. Single copy orthologs expressed in at least one out of the four samples were used for hierarchical clustering via the function hclust() from the R package stats v4.1.1, then plotted using R packages ggdendrogram and ggplot2 v3.3.5. The linear correlation of each single copy ortholog expression between the two host was calculated using ggpubr v0.4.0. The scatterplots and volcano plots were generated by ggplot2 v3.3.5.

### Comparison of gene expression in expanded gene families

The *C. elegans* and *C. briggsae* genes in PALS and DUF713 families were obtained from the output of HMMER v3.3.2 [58] using a E-value threshold of <1e-5. The protein sequences of the genes were obtained from Wormbase ParaSite (https://parasite.wormbase.org/) and aligned by M-Coffee using default settings [60, 61]. A phylogenetic tree for each family was generated using MrBayes [62, 63] with the following settings: lset nst = 1, rates = invgamma, nruns = 2, stoprule = YES, stopval = 0.05 and mcmcdiagn = YES. A dendrogram of the phylogenetic tree of each gene family was created using the function ReadDendrogram from the R package DECIPHER v2.22.0. For each gene family, the orthologs' log2 fold change values across the *C. elegans* and *C. briggsae* samples were plotted as a heatmap using ggplot2 v3.3.5.

## Results

### Gene expression of *N. parisii* and *N. ausubeli* is similar in *C. elegans* and *C. briggsae*

To understand how gene expression of related microsporidia and hosts species is conserved, we infected *Caenorhabditis* nematodes with *Nematocida* microsporidia (Fig 1A). We designed our infection experiment to compare transcriptional differences between related microsporidia and host species (Fig 1B). Unlike previous transcriptional profiling experiments of *Nematocida* infection, which were done as continuous infections at 25˚C [27, 38], we infected the worms at 21˚C for a short period of time to synchronize the infections. We pulse-infected *C. elegans* and *C. briggsae* with either *N. parisii*, *N. ausubeli*, or a mock treatment for 2.5 hours, washed to remove spores from outside the worms, and then replated the animals for a total of 10, 20, or 28 hours of infection at 21˚C. We chose these three timepoints because at 21˚C, *Nematocida* is in the sporoplasm stage at 10 hour post infection, and in the meront stage at 20 and 28 hours post infection, which allowed us to compare gene expression levels at these growth stages [26, 27] (Fig 2A). Each condition was done once, except for the 10-hour time point which was performed in duplicate (Fig 1A). Samples were stained using a probe specific to the *Nematocida* 18S rRNA and we observed that greater than 75% of each population was infected (Fig 2A and 2B). Compared to *N. parisii*-infected animals, we observed more parasite in *N. ausubeli*-infected hosts at 28 hours, consistent with a previous report that *N. ausubeli* grows faster than *N. parisii* [26]. RNA from infected and uninfected animals was extracted and sequenced. To determine the expression of microsporidia genes, we mapped reads of *N. parisii* and *N. ausubeli* to their respective genomes (S1 Table). At 10 hours post infection, the overall percentage of reads from either *N. parisii* or *N. ausubeli* were <1%, which increased to ~3–5% at 28 hours post infection (Fig 2C).

To define the transcriptional patterns of *N. parisii* and *N. ausubeli* during infection, we analysed RNA-seq results of each microsporidia species between *C. elegans* and *C. briggsae* at 10, 20, and 28 hours post infection. First, we used principal component analysis to compare microsporidia gene expression in each of these samples. We observed that at the 10-hour time

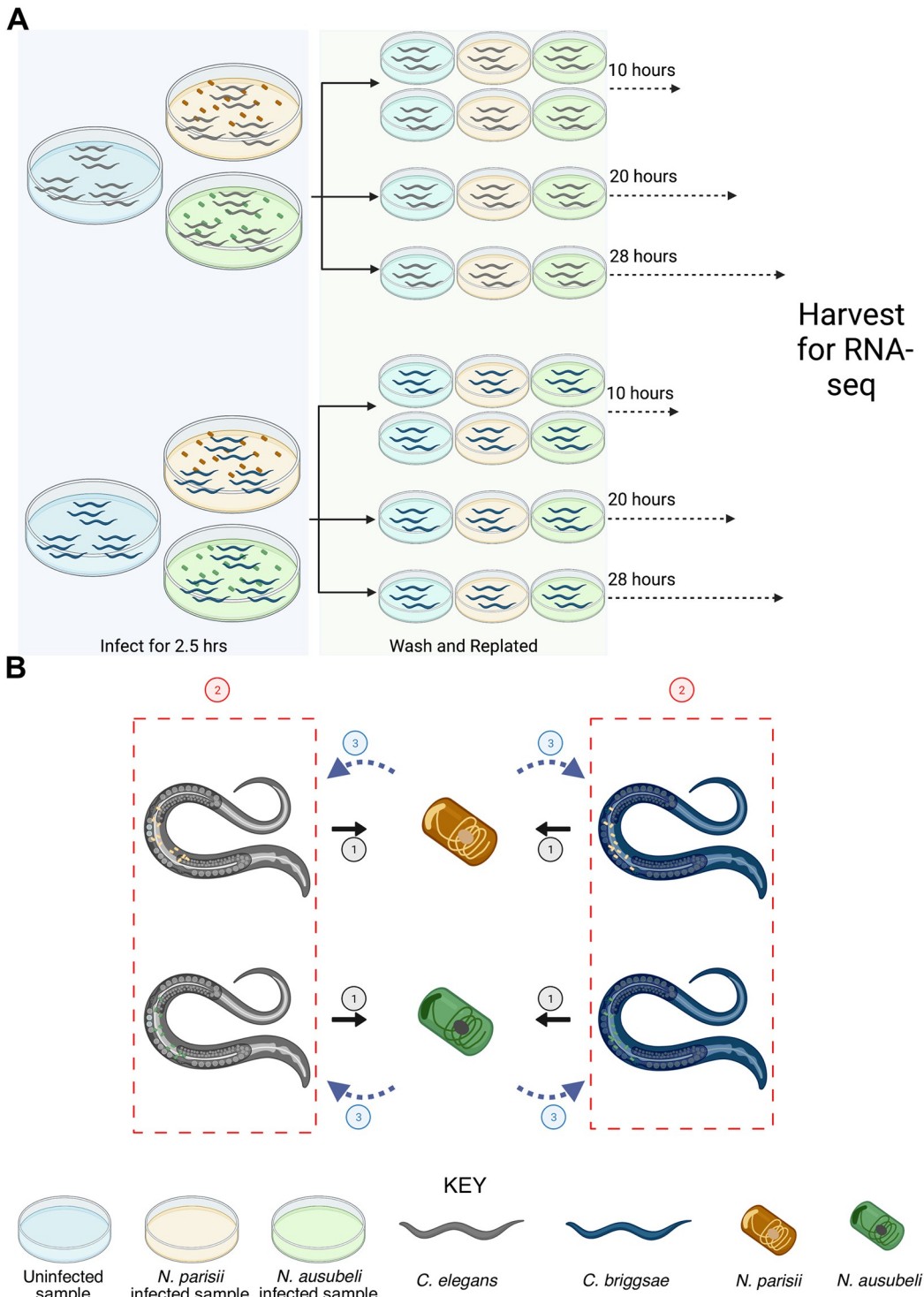

**Fig 1. Overview of study design.** (A) Schematic diagram of the RNA-seq experiment. *C. elegans* and *C. briggsae* were either not infected or infected with *N. parisii* or *N. ausubeli* separately at the L1 stage for 2.5 hours. Next, animals were washed to remove any remaining microsporidia spores and replated. Worms were harvested for RNA-sequencing at 10, 20 and 28 hours after infection. (B) The three comparisons of transcriptional responses made in this study are represented. Comparison 1 is microsporidia gene expression between hosts, comparison 2 is the response of each host species to different microsporidia species, and comparison 3 is the conserved response between different hosts to each microsporidia species. Figure was created with BioRender (www.biorender.com).

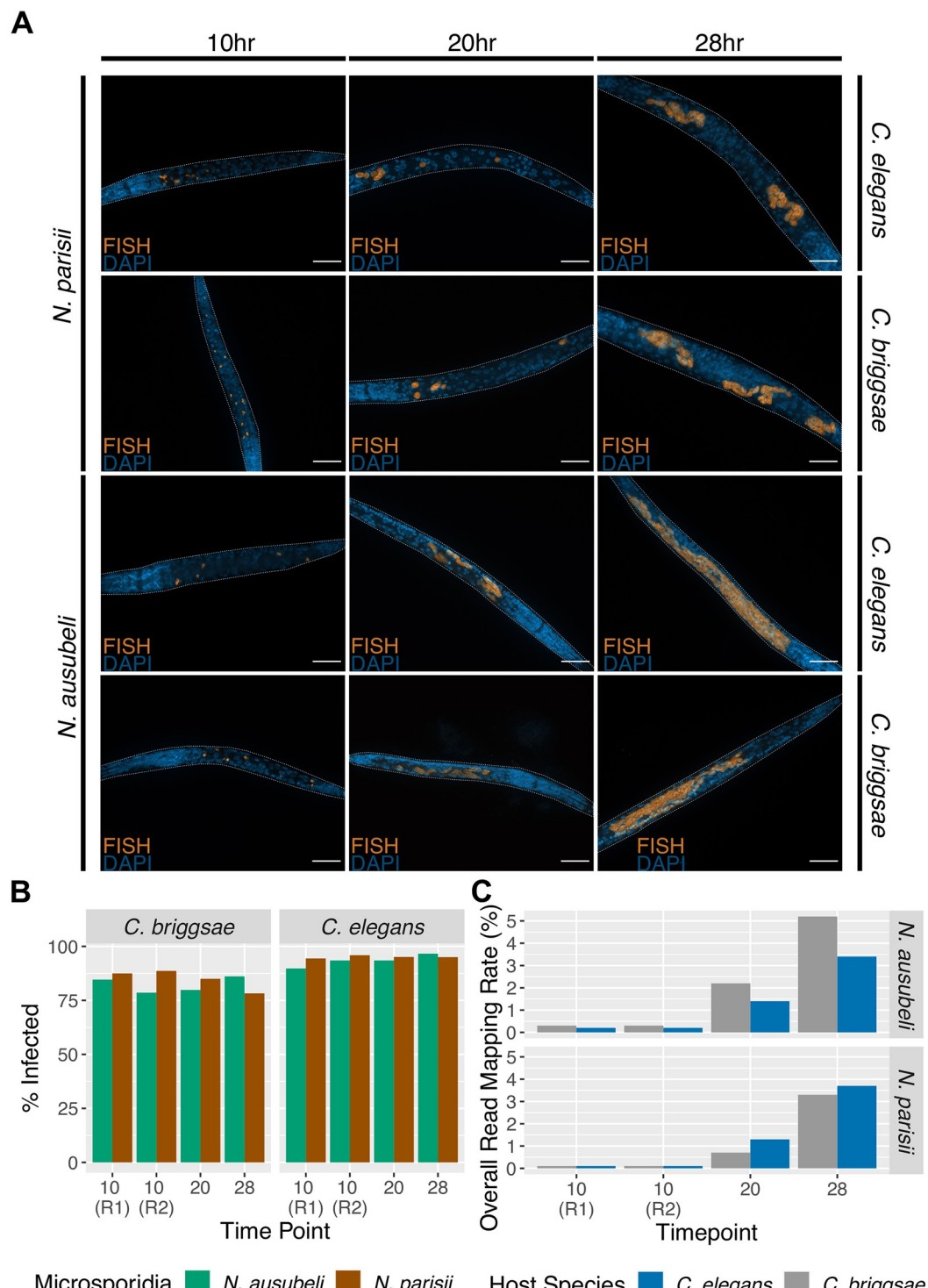

**Fig 2. *Nematocida* infection of *Caenorhabditis* hosts.** (A) Pulse-infected *C. elegans* or *C. briggsae* were fixed and then stained with DAPI to detect nuclei and a FISH probe specific to *Nematocida* 18S rRNA to detect either *N. parisii* or *N. ausubeli* at 10, 20 and 28 hours after infection. Representative images of infection are shown. Scale bars are 23 μm. (B) Percentage of infected animals at different timepoints. Between 39 to 220 animals were counted for each sample. (C) Overall read mapping rate of *Nematocida* in each host at different time points, calculated by TopHat during alignment to the respective reference genomes. R1 or R2 indicates the replicate samples at the 10-hour timepoint.

point, there is a larger difference between expression in the two hosts, and at the later time points, expression in the two hosts is similar (Fig 3A and 3B). The 20- and 28-hour time points in *N. parisii* cluster closely together, but there is a larger difference between these time points in *N. ausubeli*, likely due to the accelerated growth of this species [26]. Moreover, we observed the expression of genes in each microsporidia species to be similar between the two *Caenorhabditis* hosts across the timepoints (Fig 3C and 3D, S1A and S1B Fig). Strong correlation of *Nematocida* gene expression between *C. elegans* and *C. briggsae* was also observed, with similar levels of correlation of each microsporidia species in either host than between replicates in the same host (Fig 3E and 3F, S1C and S1D Fig). These similarities between expression in different host species also did not change across time points as the parasite had gone through larger amounts of replication. At 10 hours post infection, over 45% of *Nematocida* genes are within 2-fold of each other, and over 60% are within four-fold of each other. As the time post infection increased to 28 hours, over 75% of *Nematocida* genes are within two-fold of each other, and more than 89% are within four-fold of each other. The similarity is even more pronounced in highly expressed genes (genes with greater than 100 FPKM in either host) at the 20- and 28-hour time point where 95–99% are within four-fold of each other.

To determine genes that are significantly differentially regulated depending upon the host species infected, we compared the 10-hour time points of each *Nematocida* species between the two hosts at 10 hours post infection (Fig 4A and 4B). We identified 34 differentially regulated genes in *N. parisii* and 11 in *N. ausubeli* (alpha< 0.05) (S1 Table). Most of these differentially regulated genes had higher expression in *C. elegans* than *C. briggsae* (31/34 in *N. parisii* and 7/11 in *N. ausubeli*). Notably, differentially regulated genes were significantly enriched for ribosomal protein genes in both *N. parisii* (17/34, Fisher's exact test p-value = $2.2 \times 10^{-18}$) and *N. ausubeli* (6/11, Fisher's exact test p-value = $1.2 \times 10^{-8}$). Taken together, our results indicate that the transcriptional programs of *N. parisii* and *N. ausubeli* are largely independent of which *Caenorhabditis* hosts they infect.

## Shared and unique transcriptional responses of each *Caenorhabditis* host to *N. parisii* and *N. ausubeli* infection

Different microsporidia species may induce similar or distinct transcriptional responses in the same host. To address this question, we compared how either *C. elegans* or *C. briggsae* responded to infection with either *N. parisii* or *N. ausubeli* at the 10-hour timepoint for which we had replicate data. We first mapped the reads from each sample to the corresponding host genome (S2 Table). In *C. elegans*, we identified 875 genes that are significantly differentially regulated in *N. parisii* infected samples and 735 genes that are significantly differentially regulated in *N. ausubeli* infected samples (Fig 5A and 5B and S3 Table). For *C. briggsae*, 1091 genes are significant in *N. parisii* infected samples while 449 were significant in *N. ausubeli* infected samples (Fig 5C and 5D and S3 Table). There are more downregulated than upregulated genes in each sample. Next, we directly compared shared genes that are upregulated in each of the *Caenorhabditis* hosts when infected by *N. parisii* or *N. ausubeli*. We observed significant overlap in the transcriptional response to these infections with 222 upregulated genes shared in *C. elegans* and 117 upregulated genes shared in *C. briggsae*.

We next compared both the overlapping and unique genes between samples to determine if they are enriched in known biological functions. Statistical enrichment tests indicate that only broad categories were enriched in the unique or shared genes between samples. We observed that the shared upregulated genes in *N. parisii* and *N. ausubeli* infected *C. elegans* are enriched for GO Biological Process associated with metabolism; as well as GO Molecular Process associated with catalytic activity (FDR<0.05, S4 Table). We did not observe any enrichment in the

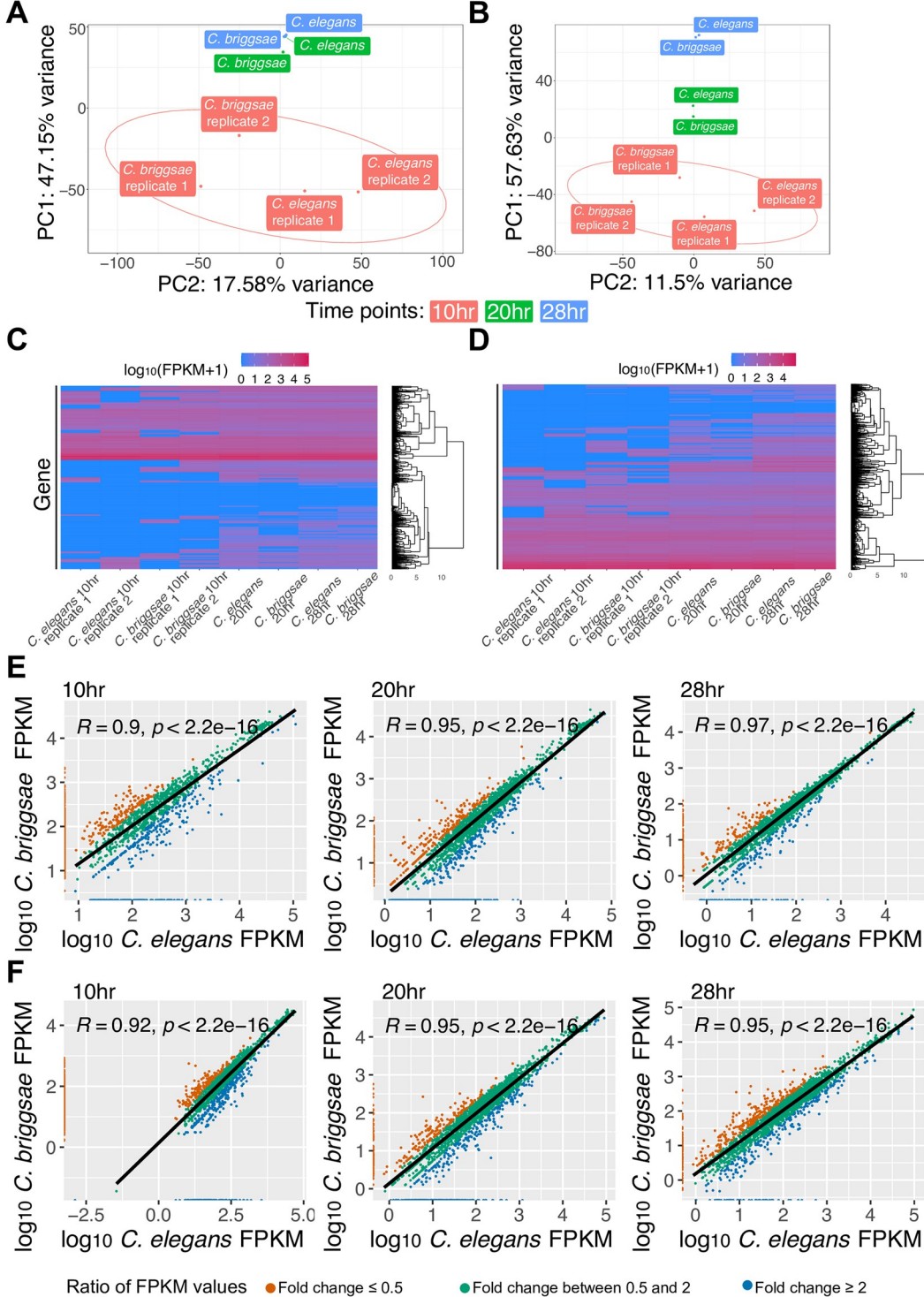

**Fig 3.** ***Nematocida* gene expression in different host species.** Principal component analysis of (A) *N. parisii* and (B) *N. ausubeli* in *C. elegans* and *C. briggsae* at 10, 20 and 28 hours. Circles represent confidence ellipses around each strain at 95% confidence interval. (C-D) Heatmap of transcriptional profiles of genes expressed in *N. parisii* (C) and *N. ausubeli* (D). Rows represent gene clustered hierarchically. Scale is of differential regulation of infected compared to uninfected samples. (E-F) Scatterplot of log10 FPKM values in *N. parisii* (E) and *N. ausubeli* (F) when infecting *C. briggsae* and *C. elegans*. Each point represents a microsporidian gene. Pearson correlation value and p-value are indicated on the top left of each plot. The ratio of FPKM values for each gene between *C. elegans* and *C. briggsae* is demonstrated by the colour of the points, which is described in the legend at the bottom.

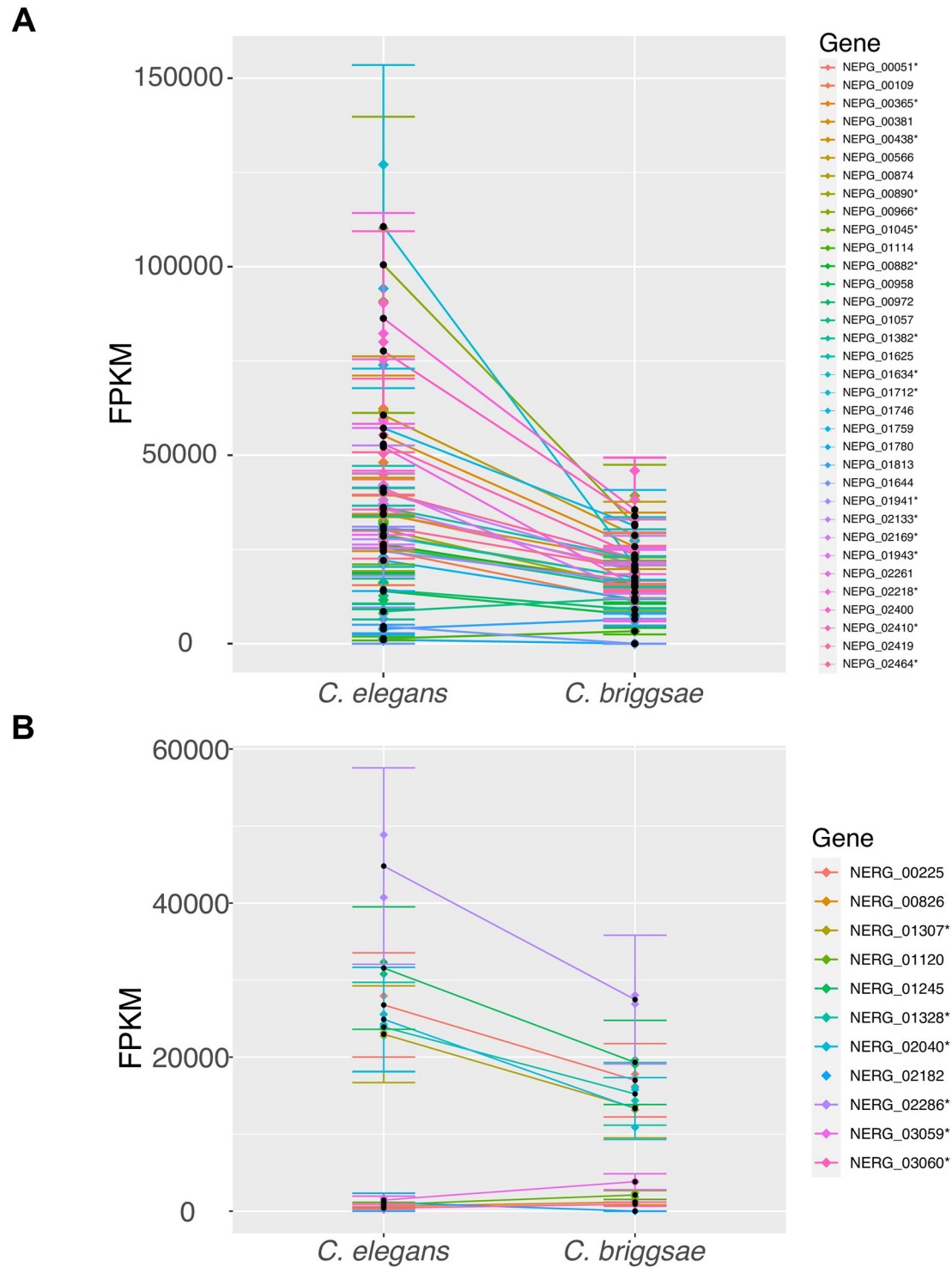

**Fig 4. Differentially regulated genes of *Nematocida*.** (A-B) Line plots of the FPKM values with confidence intervals, calculated for the differentially regulated genes (FDR<0.05) in *N. parisii* (A) and *N. ausubeli* (B) at 10 hours post infection. * indicates ribosomal protein-encoding genes. The aggregate sample value of the two replicates is noted as a black dot within the confidence interval, while the value for each replicate is shown as a dot of matching colour along the confidence interval. Error bars of each gene show the variability of the distribution of FPKM values.

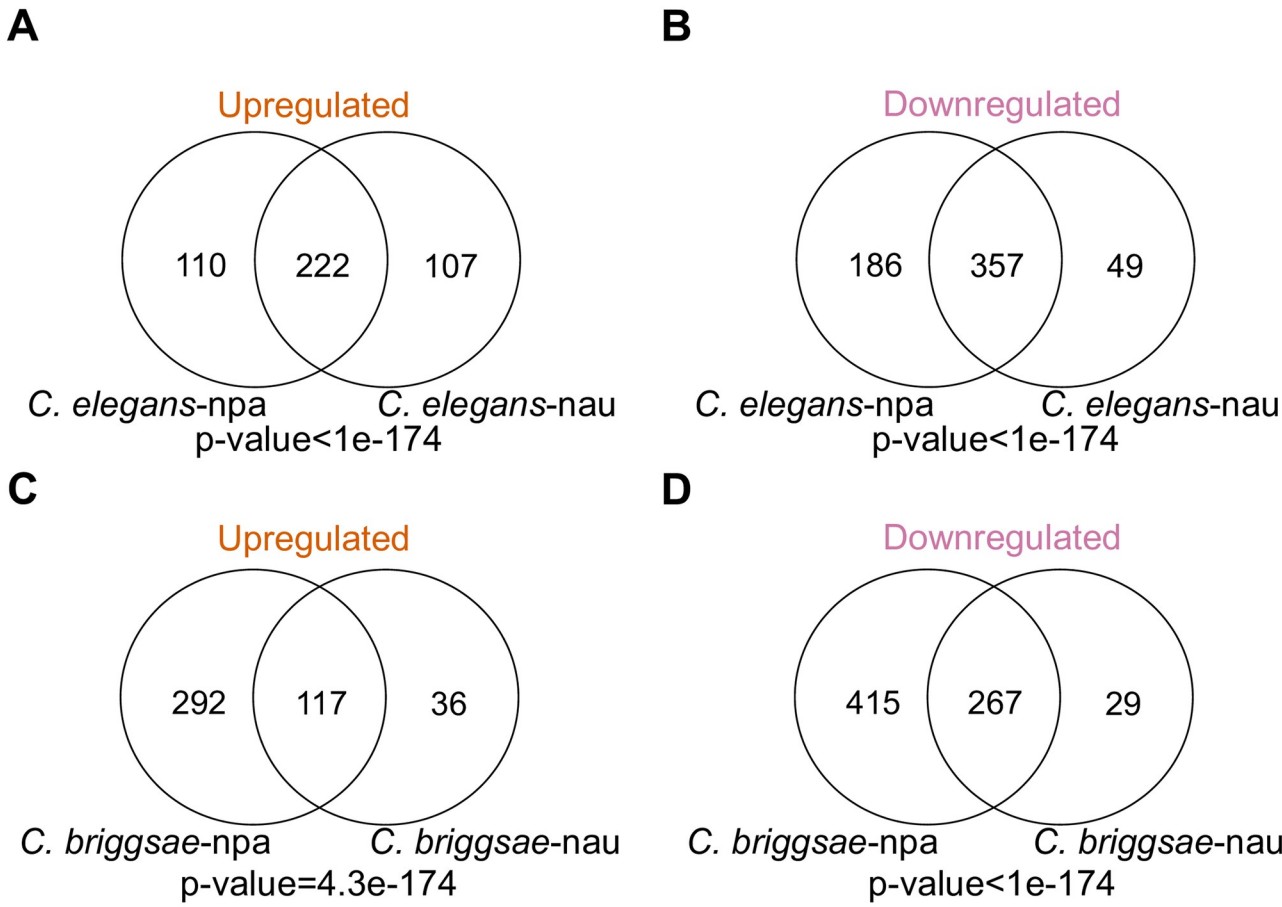

**Fig 5. Transcriptional responses of *Caenorhabditis* hosts to *Nematocida* infection.** (A-D) Venn diagrams of genes in *C. elegans* (A-B) or *C. briggsae* (C-D) that are upregulated (A and C) or downregulated (B and D) when infected by *N. parisii* or *N. ausubeli*. npa (*N. parisii)* and nau (*N. ausubeli*).

shared upregulated genes in *C. briggsae*, however the 267 shared downregulated *C. briggsae* genes showed enrichment for small molecule binding and structural constituent of cuticle (S4 Table). Of the non-shared genes, only the 415 downregulated genes in *C. briggsae* infected with *N. parisii* showed enrichment for GO Molecular Process associated with structural constituent of cuticle and protein binding (S4 Table).

We then compared our RNA-seq data to previously published studies of the transcriptional response to *N. parisii* infection (S5 Table). Bakowski *et al.* performed RNA-seq at multiple time points of a germline-deficient mutant of *C. elegans* continuously infected with spores at the L3/L4 stage at 25˚C. We compared our 10-hour time point to the five time points in the Bakowski *et al.* study and observed strong overlap especially at the 8-, 16-, and 30-hour time points (S2A Fig). We observed significant overlap for all the time points with both our *N. parisii* and *N. ausubeli* 10-hour post infection samples (S2B and S2C Fig). We also compared our data to Chen *et al.* where *C. elegans* was infected with *N. parisii* at the L3 stage at 20˚C. We observed significant overlap with both our 10-hour *N. parisii* and *N. ausubeli* data (S2D and S2E Fig). Chen *et al.* also measured Orsay virus infected *C. elegans* and Santeuil virus infected *C. briggsae* and we saw significant similarities to these as well (S2D and S2E Fig). Together our analysis demonstrates that despite the experimental differences between studies, a largely similar upregulated host response is consistently observed.

## Evolutionary conserved host response to *Nematocida* infection

To investigate the evolutionary conserved response of *Caenorhabditis* to *Nematocida* infection, we studied the expression levels of *C. elegans* and *C. briggsae* orthologs. We identified orthologs using Orthofinder [59]. Additionally, we determined the subset of orthologs that only had a single copy present in each nematode species. Of the orthogroups shared between *C. elegans* and *C. briggsae*, about 76% are single-copy orthologs (10826/14183). We first clustered the expression of these one-to-one orthologs, which shows a smaller cluster of upregulated genes and a larger cluster of downregulated genes (Fig 6A). When we compared the host response to infection between the two host species, we observed a modest but significant corelation of response to infection between *C. elegans* and *C. briggsae* (Fig 6B). We then compared the statistical significance to the fold change of differentially expressed genes from the different infection conditions (Fig 6C and 6D) and investigated which of these differentially expressed genes were one-to-one orthologs (Fig 6E and 6F). We observed that only 8–17% of the differentially upregulated genes were one-to-one orthologs. Furthermore, of the most highly expressed genes (greater than four-fold), only 0–2% were one-to-one orthologs. In contrast to less than 20% of significantly upregulated genes not having one-to-one orthologs, between 29–42% of significantly downregulated genes did.

As much of the upregulated response is in the non-single copy orthologs, we analysed these genes for commonalties between samples. We performed domain enrichment analysis to determine the types of proteins induced in the non-single copy orthologs by microsporidia infection. Previous transcriptional analysis of *C. elegans* infected with either *N. parisii* or Orsay virus has identified serval types of domains enriched in expression, such as F-box, MATH/BATH, PALS, DUF713, DUF684 and C-type lectins [27, 38]. In addition to these domains, we also included other types of domains that are enriched in the significantly upregulated and downregulated non-single copy orthologs, for a total of thirteen domains that we analysed (S6 Table). From this analysis, we found additional domains and genes enriched with greater than three domain-containing proteins in any of our samples. These identified domains are chitinase-like (*chil*) proteins, CUB and CUB-like domains, cytochrome P450, glucosyltransferase family 92, nematode cuticle collagen N-terminal domain, and UDP-glucuronosyltransferase.

The proportion of each type of domain or gene family in the upregulated and downregulated non-single copy orthologs was determined (Fig 7A and 7B). For *C. elegans* infected with *N. parisii*, we observed that ~29% of the significantly expressed non-single copy orthologs contained at least one of these thirteen domains, with genes containing PALS domains being the most common. For genes upregulated after infection with *N. ausubeli*, we observed similar trends with ~35% containing one of these domains. Overall, our results suggest that expanded gene families, especially of IPR, are upregulated during *Nematocida* infection in *Caenorhabditis* species [38, 42]. The downregulated genes showed similar domain patterns for both microsporidia species, with cuticle collagen and glycosyltransferase being the two most common domains. Similar patterns regarding types of domains in regulated genes induced by the two microsporidia species was also observed in *C. briggsae* (Fig 7B). Interestingly, cytochrome P450 genes, implicated to be involved in detoxification, are upregulated in both hosts, suggesting that the upregulated response to *Nematocida* overlaps with the metabolism and elimination of potentially toxic xenobiotics [64]. Moreover, we observe the upregulation of a few *chil* genes in *C. elegans* and *C. briggsae*, previously shown to be upregulated in response to epidermally-infecting oomycetes [37, 65].

As some of the same expanded gene families are upregulated in both *C. elegans* and *C. briggsae*, we sought to determine the relationships between expression of these families in response to infection. We first constructed gene trees of PALS and DUF713 families (S3 and S4 Figs).

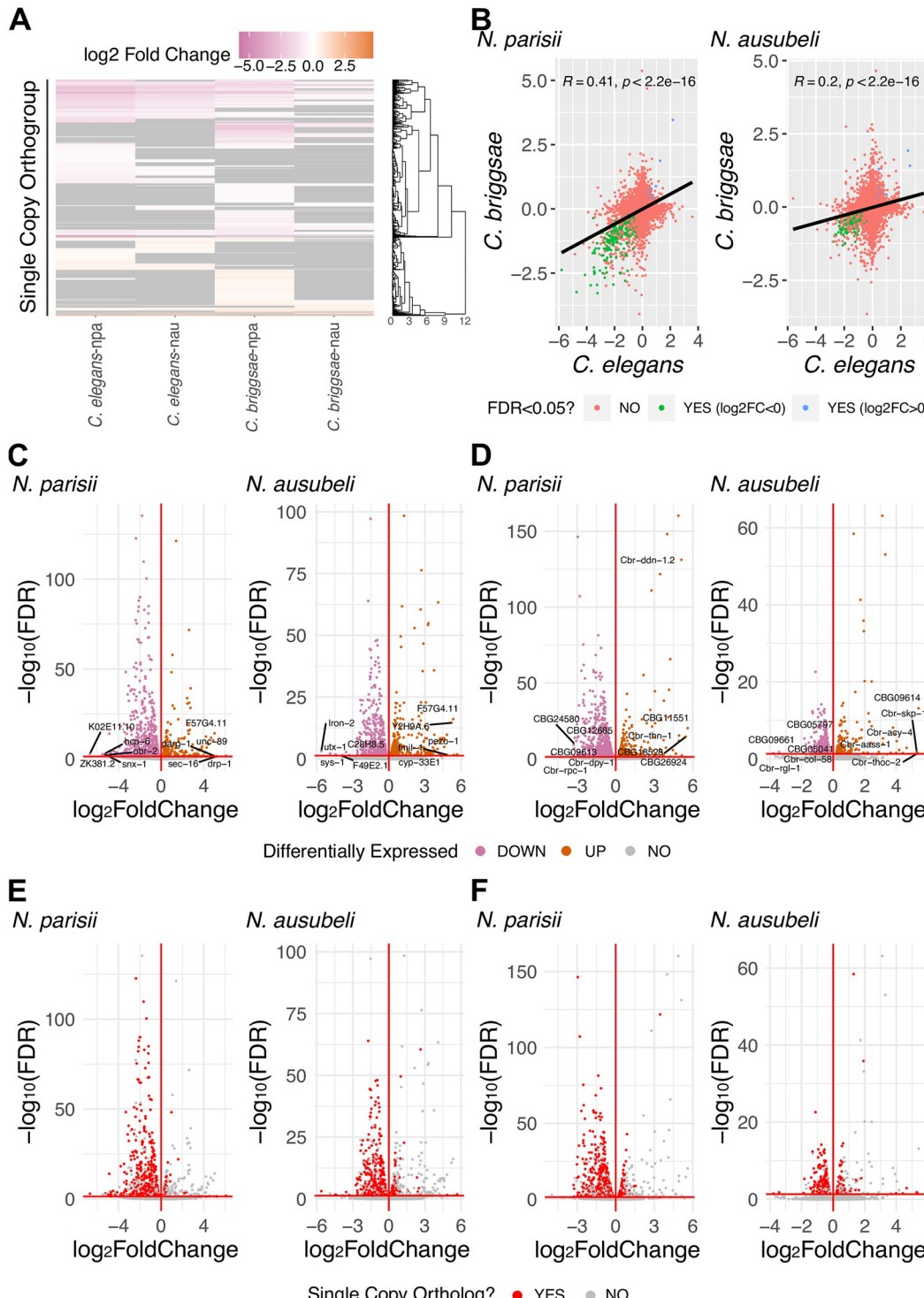

**Fig 6. Transcriptional responses of single copy orthologs in *Caenorhabditis* hosts to *Nematocida* infection.** (A) Heatmap of transcriptional profiles of single-copy orthologs in *C. elegans* and *C. briggsae* after infection by *N. parisii* or *N. ausubeli*. Rows represent genes clustered hierarchically. Scale is of differential regulation of infected compared to uninfected samples. (B) Scatterplot of log2 fold change values of single-copy orthologs expressed in *C. elegans* and *C. briggsae*. Each point represents a single-copy ortholog between the two host species. Pearson correlation values and p-values are indicated on top left of each plot. The ratio of log2 fold change values for each single copy ortholog between *C. elegans* and *C. briggsae* is demonstrated by the colour of the points. (C-D) Volcano plots of genes expressed in *Nematocida* infected *C. elegans* (C) and *C. briggsae* (D). Upregulated genes are represented with orange points; downregulated genes are represented with pink

points. The top 5 upregulated and downregulated genes are labelled. (E-F) Volcano plots of genes expressed in *Nematocida* infected *C. elegans* (E) and *C. briggsae* (F). Single-copy orthologs are represented with red points and non-single copy orthologs represented with grey points. npa (*N. parisii*) and nau (*N. ausubeli*).

Next, we compared the expression levels of these orthologs in our 10-hour samples. This analysis shows that some phylogenetically related PALS and DUF713 genes are upregulated in both *C. elegans* and *C. briggsae* in response to *Nematocida* infection (Fig 8A and 8B).

## Discussion

To understand the conservation of transcriptional responses during microsporidia infection, we took the approach of comparing interactions between related microsporidia and host species. We found that responses between *Caenorhabditis* and *Nematocida* species are largely conserved. However, there are several limitations to our study. Although we monitored infection at multiple time points, we only performed duplicates at 10 hours post infection. Although a small number of replicates limits the ability to detect differentially regulated genes [66], we believe that our conclusions are not simply a consequence of the lack of statistical power for the following reasons. Firstly, we saw a large overlap in significantly regulated genes compared to previously published transcriptional datasets of *N. parisii* infected *C. elegans* [13, 14]. The conclusions we reach in our study also agree with those from another transcriptional response comparison from our group that generated RNA-seq data in triplicate of *C. elegans* infected at a latter timepoint with four different species of microsporidia, including *N. parisii* and *N. ausubeli* [67]. Secondly, previous statistical power calculations for RNA-seq analysis over a range of fold-changes and number of replicates suggests that a small number of replicates is sufficient to capture highly-expressed genes, such as the highly expressed IPR genes observed in our study [66, 68, 69]. Finally, previously published transcriptional datasets from Bakowski *et al.*, (2014) and Chen *et al.*, (2017), which we compared our dataset to, were also performed in duplicates or triplicates. Another limitation is that our study only analysed two pairs of hosts and microsporidia species. Additionally, differences in transcriptional responses between species could be dependent upon particular host and microsporidia strains or environmental conditions.

Our results suggest that *Nematocida* microsporidia species mostly do not sense and respond differently depending upon their host environment. The main difference we observed in *Nematocida* gene expression between the two hosts was an upregulation of both small and large ribosomal subunit genes in *C. elegans* at 10 hours post infection. We also observed about three-fold more ribosomal genes being differentially regulated in *N. parisii* than *N. ausubeli*. After invasion both species undergo a lag period before starting replication. This lag period is slightly shorter in *N. ausubeli* and the larger increase of ribosomal genes in *N. parisii* is potentially related to these differences in lag time [26]. The lack of large differences in the host response also suggests that these *Nematocida* microsporidia are likely not differentially regulating host genes for their own benefit.

How hosts sense microsporidia infection is not fully understood. In mammals, several toll-like receptors are necessary for immune activation [70], but the proteins that *C. elegans* uses to sense and respond to microsporidia infection are mostly unknown. Several negative regulators of the IPR are known, including PALS-22, LIN-35, and PNP-1, but none of these are known to be necessary for the response to infection [28, 31–33]. Recently, a basic-region leucine-zipper transcription factor, ZIP-1, was found to positively regulate a subset of IPR genes [71]. A viral RNA receptor, DRH-1, is necessary for activation of the IPR by the Orsay virus, but not for microsporidia infection [72]. Whether there is an equivalent protein that is specifically

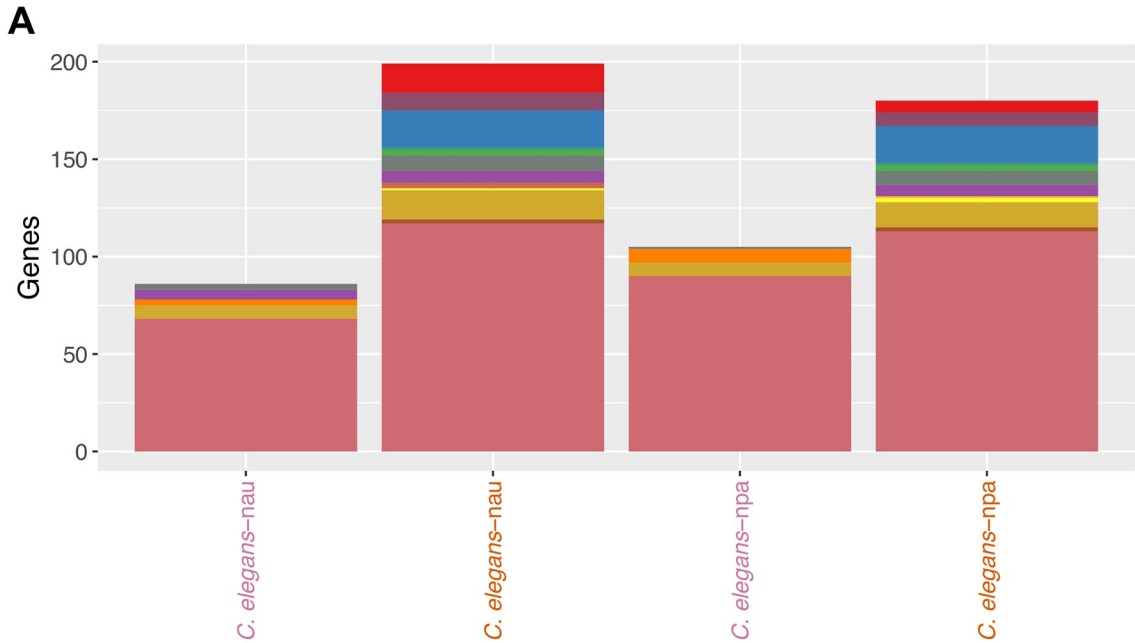

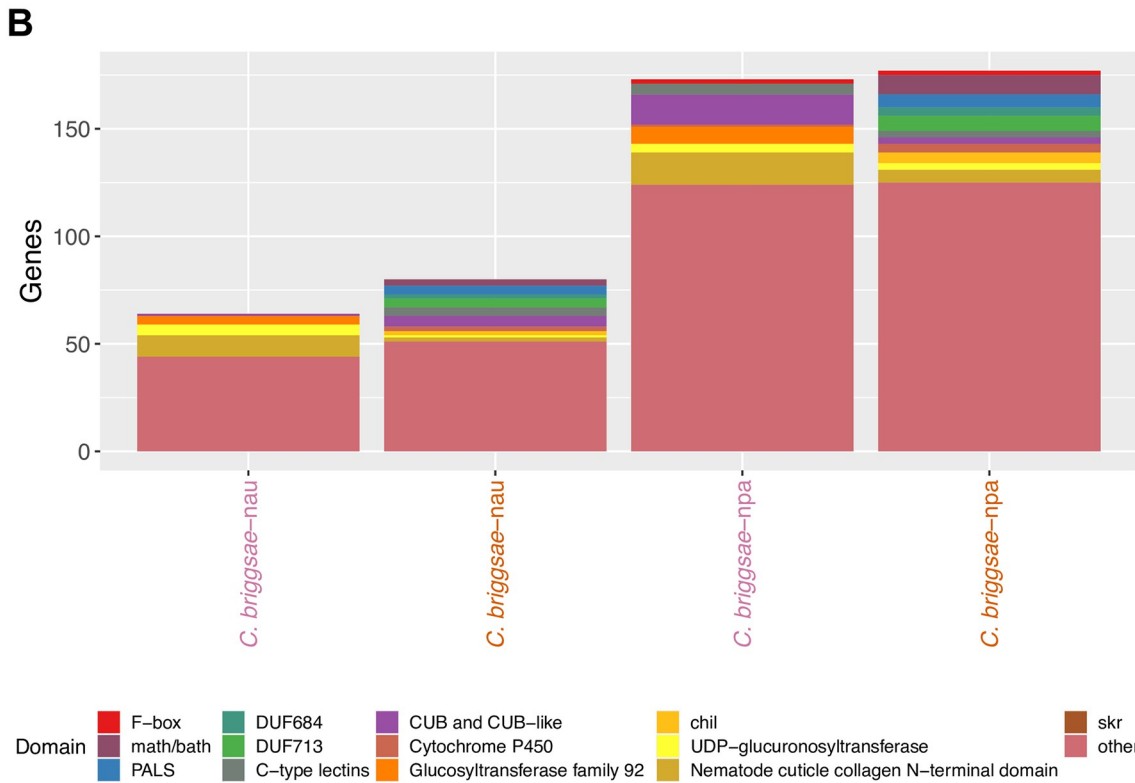

**Fig 7. Transcriptional responses of non-single copy orthologs in *Caenorhabditis* hosts to *Nematocida* infection.** Domain enrichment analysis of significantly upregulated and downregulated non-single copy ortholog genes in *C. elegans* (A) and *C. briggsae* (B). npa (*N. parisii*) and nau (*N. ausubeli*). Upregulated samples are represented with orange text; downregulated samples are represented with pink text.

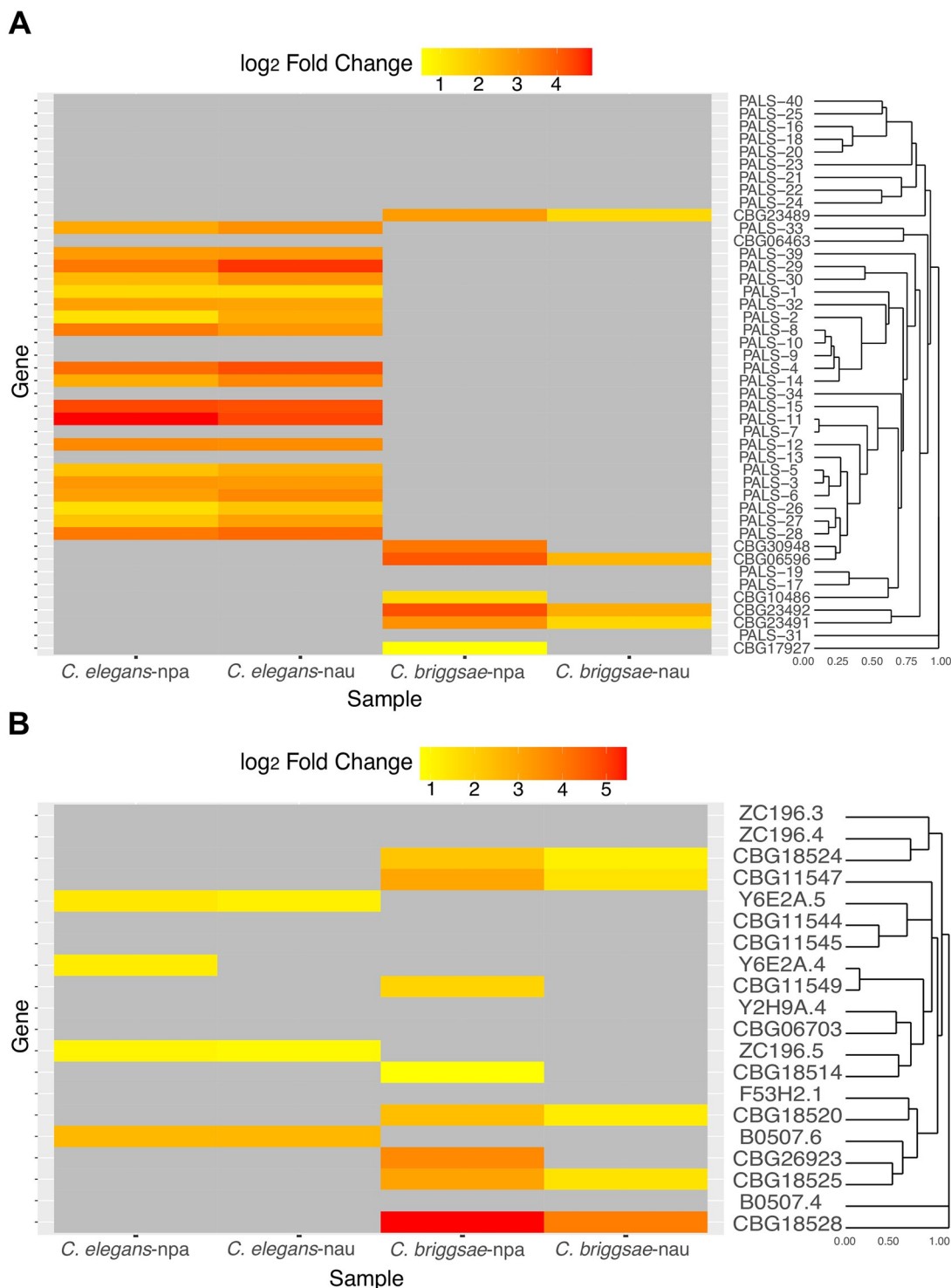

**Fig 8. Conserved response of *Caenorhabditis* hosts to *Nematocida* infection.** (A-B) Heatmap of transcriptional profiles of differentially regulated genes (FDR<0.05) in the PALS (A) and DUF713 (B) gene families. Scale is of differential regulation of infected compared to uninfected samples. npa (*N. parisii)* and nau (*N. ausubeli*).

involved in detecting microsporidia infection in *C. elegans* is unknown. Although a previous study showed that three IPR genes were induced to a lesser extent by *N. ausubeli* than *N. parisii* [13], by comparing the full transcriptional profile we see a largely similar set of genes being induced. As these different microsporidia species induce a similar transcriptional response, this suggests that some common feature, such as invasion, is detected by the host.

Nematodes, insects, and vertebrates all have strong transcriptional responses to infection by microsporidia, though the responses appear to be quite diverse [73, 74]. For example, anti-microbial peptides are observed to be upregulated in silkworms and cytokines are induced in infected human cells [75, 76]. The responses seen in these other animals are quite different than what is observed in *Caenorhabditis*. One reason for this is that many IPR genes are part of large gene families, such as PALS [30], that are not conserved outside of *Caenorhabditis*. Little is known about how a host responds to different species of microsporidia. A study examining two species of microsporidia that infect the same mosquito found that a horizontal transmitted species elicited a strong transcriptional immune response, but this response was not enriched during infection with a vertically transmitted species [77]. Further studies will be necessary to know if the similar immune responses we observe to infection by two horizontally transmitted microsporidia species are common in other hosts infected by different microsporidia.

Microsporidia that infect free-living terrestrial nematodes appear to be common, and there are other genera and families of nematodes that can be cultured in the laboratory and infected with microsporidia. Additionally, there are other genera of microsporidia that have been shown to infect *C. elegans* and other nematodes. Some nematode-infecting microsporidia also infect multiple genera of nematodes, which could facilitate cross-species host expression [13]. These types of broader comparisons would allow for a fuller view of how animals evolve transcriptional responses to microsporidia infection.

## Supporting information

**S1 Fig. Transcriptional response of *Nematocida* species in *Caenorhabditis* hosts.** (A-B) Scatterplot of *N. parisii (A)* and *N. ausubeli (B)* log10 FPKM values between *C. briggsae* and *C. elegans* replicates at 10 hours. Pearson correlation values and p-values are indicated on the top left of each plot. The ratio of FPKM values for each gene between the replicates is demonstrated by the colour of the points. (C-D) Gene expression pattern of *N. parisii* (C) and *N. ausubeli* (D) in *C. elegans* and *C. briggsae*.
(TIF)

**S2 Fig. Data from this study has high overlap with previously published intracellular infection samples.** (A) Heatmap of transcriptional profiles of differentially regulated genes (FDR<0.05) across five samples from Bakowski *et al.* and our 10hr *N. parisii* and *N. ausubeli* infected *C. elegans* samples. Only genes expressed in at least four out of the eight total samples are included. (B-C) Statistical overlap of significant genes between respective *N. parisii* or *N. ausubeli* infected *C. elegans* samples and Bakowski *et al.* samples across five timepoints. (D-E) Statistical overlap of significant genes between *N. parisii* (D) or *N. ausubeli* (E) infected *C. elegans* samples and Chen *et al.* Orsay virus infected *C. elegans* sample at 12 hours. (F) Statistical overlap of significant genes between respective *N. parisii* or *N. ausubeli* infected *C. briggsae* sample and Chen *et al.* Santeuil virus infected *C. briggsae* sample at 12 hours. The p-value of each comparison is indicated on top of each Venn diagram. npa (*N. parisii)* and nau (*N. ausubeli*).
(TIF)

**S3 Fig. Phylogram of *C. elegans* and *C. briggsae* PALS family genes.** Genes with the same name that end with different letters indicate protein isoforms. Node values indicate posterior

probabilities for each split as a percentage. The scale bar indicates average branch length measured in expected substitutions per site.
(TIF)

**S4 Fig. Phylogram of *C. elegans* and *C. briggsae* DUF713 family genes.** Genes with the same name that end with different letters indicate protein isoforms. Node values indicate posterior probabilities for each split as a percentage. The scale bar indicates average branch length measured in expected substitutions per site.
(TIF)

**S5 Fig.**
(TIF)

**S1 Table. FPKM values of gene expression and differentially expressed microsporidia genes.**
(XLSX)

**S2 Table. Normalised RNA-seq counts of *C. elegans* and *C. briggsae* for the transcriptional analysis generated by Alaska.**
(XLSX)

**S3 Table. Differentially expressed genes of *C. elegans* and *C. briggsae* exposed to the two species of microsporidia.**
(XLSX)

**S4 Table. PANTHER GO enrichment analyses.** Sheet S1 contains results of the statistical enrichment tests of the 222 significantly upregulated genes shared between *N. parisii* and *N. ausubeli* infected *C. elegans*. Sheet S2 contains results of the statistical enrichment tests of the 267 significantly downregulated genes shared between *N. parisii* and *N. ausubeli* infected *C. briggsae*. Sheet S3 contains results of the statistical enrichment tests of the 415 significantly downregulated genes specifically in *N. parisii* infected *C. briggsae*.
(XLSX)

**S5 Table. List of gene set overlaps with previously published infection samples.**
(XLSX)

**S6 Table. Gene classes and domains used for enrichment analyses.**
(DOCX)

## Acknowledgments

We thank Hala Tamim El Jarkass, Meng A. Xiao, and Winnie Zhao for providing helpful comments on the manuscript. We thank Winnie Zhao for technical help for the comparisons to the Bakowski data. Some strains were provided by the CGC and we thank WormBase.

## Author Contributions

**Conceptualization:** Yin Chen Wan, Aaron W. Reinke.

**Data curation:** Aaron W. Reinke.

**Formal analysis:** Yin Chen Wan.

**Funding acquisition:** Emily R. Troemel, Aaron W. Reinke.

**Investigation:** Yin Chen Wan, Aaron W. Reinke.

**Methodology:** Aaron W. Reinke.

**Project administration:** Aaron W. Reinke.

**Supervision:** Aaron W. Reinke.

**Visualization:** Yin Chen Wan.

**Writing – original draft:** Yin Chen Wan, Aaron W. Reinke.

**Writing – review & editing:** Yin Chen Wan, Emily R. Troemel, Aaron W. Reinke.

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
