## [Decision Letter · Decision Letter 0]

19 Sep 2022

PONE-D-22-23486Conservation of Nematocida microsporidia gene expression and host response in Caenorhabditis nematodesPLOS ONE

Dear Dr. Reinke,

Thank you for submitting your manuscript to PLOS ONE. After careful consideration, we feel that it has merit but does not fully meet PLOS ONE’s publication criteria as it currently stands. Therefore, we invite you to submit a revised version of the manuscript that addresses the points raised during the review process.

We look forward to receiving your revised manuscript.

Kind regards,

Nicolas Corradi

Academic Editor

PLOS ONE

Additional Editor Comments :

Dear Authors.

Your submission has been reviewed by two referee who has judged your work to be publishable in PloS One pending minor changes.

I agree with the referees, and I look forward to receiving a revised version of your work.

Best wishes, Nicolas Corradi

Reviewers' comments:

Reviewer's Responses to Questions

**Comments to the Author**

1. Is the manuscript technically sound, and do the data support the conclusions?

Reviewer #1: Yes

Reviewer #2: Yes

2. Has the statistical analysis been performed appropriately and rigorously? 

Reviewer #1: I Don't Know

Reviewer #2: Yes

3. Have the authors made all data underlying the findings in their manuscript fully available?

Reviewer #1: Yes

Reviewer #2: Yes

4. Is the manuscript presented in an intelligible fashion and written in standard English?

Reviewer #1: Yes

Reviewer #2: Yes

5. Review Comments to the Author

Reviewer #1: In this manuscript, the authors used RNAseq data to investigate if distinct microsporidia species from the genus Nematocida (N. parisii and N. ausubeli) express different genes when infecting different nematode hosts from the genus Caenorhabditis (C. elegans and C. briggsae). They also investigated if these nematodes hosts differ in their response to the Nematocida spp. infections. From their findings, the authors conclude that neither the pathogens nor the hosts express different genes depending on which species is infecting or being infected, respectively.

Overall, I found the manuscript to be easy to read and did not detect any major flaw (my comments are described below). I believe that the manuscript will be suitable for publication in PLOS one pending minor revisions.

# 1. The authors spent a lot of effort investigating potential expression differences between two species of Nematocida infecting two species of Caenorhabditis and the overall take home message of their manuscript is that they found no major difference. However, one of the main questions that came to my mind while reading this manuscript is: Are those species really that different? Because if not, then one should not expect any major difference in gene expression levels.

In the introduction (lines 46-61) the authors mention that N. ausubeli grows faster than the other microsporidian species. Likewise, the authors state that the two nematodes live in distinct but overlapping locales but that doesn’t tell us much about their biological differences, if any. What else makes these species distinct? I am neither a Caenorhabditis nor a Nematocida expert and, overall, I felt that I was missing some crucial information to properly appreciate the authors findings while going through the manuscript. Although the authors somewhat address this question later on at the start of their discussion (on page 5, lines 249-254) by talking about amino acid differences (are those significant?), this should be clearly addressed in the introduction.

In the same trail of thought, were the nematode host species C. elegans and C. briggsae selected because they showed distinct responses to other fungal pathogens? The authors succinctly mention nematode responses to fungal infections in their introduction and instead focus on their response to bacterial infections (lines 75-88). However, microsporidia are fungi (or at least closely related) and expanding on nematode responses to fungal infections would be more informative than using bacteria as a frame of reference.

Other minor comments:

# 2. I’m assuming that the 10, 20 and 28h post-infection time points for RNAseq are relevant to the microsporidia biology but it would be informative to have a brief statement explaining why these time points were selected. Was it based on the infection progression as observed in the FISH results from Fig 2A?

# 3. Page 4 lines. 175-178. 'Using statistical enrichment tests, we observed that the shared upregulated genes in N. parisii and N. ausubeli infected C. elegans are enriched for GO Biological Process associated with metabolism; as well as GO Molecular Process associated with catalytic activity (FDR<0.05, Supplementary table S4)'.

This statement in uninformative. Metabolism and catalytic activity are incredibly broad categories. Growing organisms have metabolic/catabolic activity. That is a given. Is there any gene ontology term other than these vague categories that stands out?

# 4. Page 7. RNA sequencing. Which library kit was used to prepare the sequencing libraries? The authors state that mRNA libraries were generated so a rRNA depletion step was likely not required but how were those mRNA libraries generated? Different kits can have different biases and this is also important to know which adapters might be present in the sequencing datasets.

# 5. Page 7. Microsporidia gene expression analysis. Were the adapters properly removed from the sequencing datasets prior to the analyses? The authors used two different methods to analyze RNAseq data, one for the microsporidia datasets and one for the nematode datasets (using the WormBase Alaska pipeline). The latter mention some sort of quality control (QC) but the first one does not mention any such QC step.

# 6. Page 7 lines 366-367: 'The paired end reads of each sample 10 hours post infection sample were submitted to Alaska v1.7.2 (http://alaska.caltech.edu)'.

What about the 20 and 28 PI samples? Were they also analyzed with the Alaska pipeline from WormBase? Did the authors mean: The paired end reads of each sample were submitted to Alaska v1.7.2 (http://alaska.caltech.edu)? The authors should also explicitly state what Alaska is. I had to reread that sentence twice as I found it confusing.

# 7. Page 7 methods and page 12 Fig 2C. From which RNAseq analysis method were the mapping metrics in Fig 2C derived; Alaska, the microsporidia one, both? A subtraction would not do here if contamination from a third party was present. Also, while I understand using the WormBase pipeline to analyze the Caenorhabditis data, this pipeline uses completely different tools than the ones used to analyze microsporidia data. Did the authors check for congruency between the two approaches?

# 8. Page 8. Lines 420 - 424. 'From this analysis, we found additional domains and genes enriched with greater than three domain-containing proteins in any of our samples. These identified domains are chitinase-like (chil) proteins, CUB and CUB-like domains, cytochrome P450, glucosyltransferase family, nematode cuticle collagen N terminal domain, and UDP-glucuronosyltransferase'.

These sentences read like results rather than methods. Also, P450 cytochromes are involved in the detoxification of xenobiotics in insects. If those were enriched/overexpressed, something important might be going on here. Microsporidia spores are also made of chitin so chitinases appear highly relevant here.

# 9. On page 1, lines 50-51. 'These two species of Nematocida are also commonly found to infect Caenorhabditis briggsae, which has been developed as a comparative species to C. elegans (Stein et al., 2003)'.

Weird phrasing: species do not arise to be compared to something and Caenorhabditis briggsae was not developed/created in lab (or was it?). I’m assuming that the authors meant that this species was used as an alternate model to facilitate comparative analyses or something like that…

# 10. On page 2, lines 83-85 'A study looking at different species of bacterial infection in the nematode Pristionchus pacificus also showed both similarities and differences between the transcriptional responses in the two hosts (Sinha et al., 2012)'.

'showed both similarities and differences' This is vague. Minor/Major differences? A few but crucial ones?

# Figures 3 and 4 legends. Replace Microsporidia by Nematocida in the titles. These results are not representative of the whole group, and they would likely differ in non-Caenorhabditis hosts.

# 12. Formatting: References are not formatted according to PLOS guidelines.

Reviewer #2: The manuscript by Wan, Troemel and Reinke characterizes the microsporidian and host transcriptomes of two Nematocida spp. infecting two Caenorhabditis spp. The central questions posed by the authors are: 1. whether the parasites change their gene expression profiles depending on the host infected, and 2. whether the host gene expression profiles differ when infected by either Nematocida spp. The conclusion is that parasite gene expression profiles are very similar, no matter what host is being used, and that the same could be said for the host as well.

I think it is an interesting paper, well written and with very nice illustrations, which is worth of publication.

I have three questions to the authors that may (or may not) be explored in order to build an improved version of the manuscript.

1) Hosts were experimentally infected and RNA was extracted at 10, 20 and 28 hours post infection, with one replicate only for the 10 hpi timepoint. On line 257 of the discussion the authors recognize that the lack of replications could have limited their ability to detect differentially expressed genes. It would be interesting to know the extent by which their conclusions have been biased by the lack of replicates. Is it possible to estimate that in any way (e.g. by generating pseudo replicates)? Alternatively, would it be possible to provide at least one good argument to the reader giving support for their claims, i.e., that the conclusions are not simply a consequence of the lack of statistical power?

2) Another aspect that needs clarification is the statement that expression profiles are “evolutionarily conserved”. I am not sure whether you could say that for such closely related species, but the reader should at least know what evolutionary scale is actually being assessed in the study. Somewhere it should be mentioned to what degree both microsporidia and both hosts genetically diverge from each other (for any standard marker such as rRNA or other phylogenetic marker gene).

3) I am wondering whether it wouldn’t be possible, with the data at hand, to test other hypotheses. It would be interesting to know, for instance, what are the differences in expression 1. between parasites independently of the host, or 2. between hosts independently of the parasite. The first is going to be an unavoidable curiosity of the manuscript reader, considering that N. parisii and N. ausubeli differ in their infection phenotypes (lines 60-61).

6. PLOS authors have the option to publish the peer review history of their article (what does this mean?). If published, this will include your full peer review and any attached files.

Reviewer #1: No

Reviewer #2: No

---

## [Author Response · Author response to Decision Letter 0]

29 Nov 2022

We thank the reviewers for their comments. We have provided a revised manuscript (where all changes are tracked) to address the reviewers’ concerns. Points raised by the reviewers have been incorporated into the manuscript and we have provided a point-by-point response to each of the reviewer’s comments: 

Reviewer #1: In this manuscript, the authors used RNAseq data to investigate if distinct microsporidia species from the genus Nematocida (N. parisii and N. ausubeli) express different genes when infecting different nematode hosts from the genus Caenorhabditis (C. elegans and C. briggsae). They also investigated if these nematodes hosts differ in their response to the Nematocida spp. infections. From their findings, the authors conclude that neither the pathogens nor the hosts express different genes depending on which species is infecting or being infected, respectively.

Overall, I found the manuscript to be easy to read and did not detect any major flaw (my comments are described below). I believe that the manuscript will be suitable for publication in PLOS one pending minor revisions.

We thank the reviewer for their kind comments.

# 1. The authors spent a lot of effort investigating potential expression differences between two species of Nematocida infecting two species of Caenorhabditis and the overall take home message of their manuscript is that they found no major difference. However, one of the main questions that came to my mind while reading this manuscript is: Are those species really that different? Because if not, then one should not expect any major difference in gene expression levels.

To address this comment, we have added additional information in the following paragraphs to the introduction:

Lines 62-69:

There are some similarities in the infections caused by these two Nematocida species; they both exclusively infect the intestine, they infect C. elegans and C. briggsae, they cause intestinal cells to fuse, and have similar life cycles (Balla et al., 2016; Zhang et al., 2016; Wadi et al., 2022). Both species also use similar types of secreted and membrane bound proteins to interface with host proteins (Reinke et al., 2017). Although the divergence time between N. parisii and N. ausubeli is unknown, the two Nematocida species share 68.3% amino acid identity (Cuomo et al., 2012; Luallen et al., 2016). These two species also have distinct growth and phenotypic characteristics during infection in C. elegans; specifically N. ausubeli displays faster growth and increased impairment of host fitness (Balla et al., 2016).

Lines 107-110:

For the hosts, we used C. elegans and C. briggsae, which diverged ~18 million years ago (Cutter, 2008). About 60% of the genes from these species are orthologous and the median percent identity between these proteins is 80%, which is approximately the same extent of divergence between humans and mice (Stein et al., 2003).

In the introduction (lines 46-61) the authors mention that N. ausubeli grows faster than the other microsporidian species. Likewise, the authors state that the two nematodes live in distinct but overlapping locales but that doesn’t tell us much about their biological differences, if any. What else makes these species distinct? I am neither a Caenorhabditis nor a Nematocida expert and, overall, I felt that I was missing some crucial information to properly appreciate the authors findings while going through the manuscript. Although the authors somewhat address this question later on at the start of their discussion (on page 5, lines 249-254) by talking about amino acid differences (are those significant?), this should be clearly addressed in the introduction.

As described above, we have modified the introduction to better clarify the differences and similarities between the Caenorhabditis and Nematocida species in lines 44-69 and 106-111.

In the same trail of thought, were the nematode host species C. elegans and C. briggsae selected because they showed distinct responses to other fungal pathogens? The authors succinctly mention nematode responses to fungal infections in their introduction and instead focus on their response to bacterial infections (lines 75-88). However, microsporidia are fungi (or at least closely related) and expanding on nematode responses to fungal infections would be more informative than using bacteria as a frame of reference.

C. elegans and C. briggsae were chosen because these two host species are infected efficiently by N. parisii and N. ausubeli under laboratory setting, similar to their natural interactions in the wild. Thus these host-parasite pairs are an ideal set of species to study transcriptional responses to microsporidia in related species. We have added additional information to clarify why we choose these species in lines 44-69 and 106-111.

Although microsporidia are early-diverging fungi, Caenorhabditis nematodes have distinct responses to Nematocida compared to other fungi. Transcriptomic response of different nematodes to pathogenic bacteria is more studied, therefore we included bacteria as a frame of reference. For clarity, we have differentiated the response to different pathogens by integrating the following text into the introduction:

Lines 80-87

In response to microsporidia infection, hosts often display large transcriptional changes. The gene expression of C. elegans in response to N. parisii infection is reported to be distinct from responses to bacteria, but similar to nodavirus infection (Bakowski et al., 2014). Although the response to N. parisii infection also shared some similarities to the response against fungi such as Drechmeria coniospora and Harposporium, the genes upregulated during N. parisii infection are still mostly specific to either microsporidia or viral infection (Bakowski et al., 2014).

Other minor comments:

# 2. I’m assuming that the 10, 20 and 28h post-infection time points for RNAseq are relevant to the microsporidia biology but it would be informative to have a brief statement explaining why these time points were selected. Was it based on the infection progression as observed in the FISH results from Fig 2A?

Yes, to address this, we have added the following text to the results: 

Line 285-288

We chose these three timepoints because at 21°C, Nematocida is in the sporoplasm stage at 10 hour post infection, and in the meront stage at 20 and 28 hours post infection, which allowed us to compare gene expression levels at these growth stages (Bakowski et al., 2014; Balla et al., 2016) (Fig 2A).

# 3. Page 4 lines. 175-178. 'Using statistical enrichment tests, we observed that the shared upregulated genes in N. parisii and N. ausubeli infected C. elegans are enriched for GO Biological Process associated with metabolism; as well as GO Molecular Process associated with catalytic activity (FDR<0.05, Supplementary table S4)'.

This statement in uninformative. Metabolism and catalytic activity are incredibly broad categories. Growing organisms have metabolic/catabolic activity. That is a given. Is there any gene ontology term other than these vague categories that stands out?

We agree with the reviewers that these are vague categories, but they are the ones that are enriched. We have modified the text in the relevant paragraph to clarify that only these broad categories were enriched.

Line 351-352

“Statistical enrichment tests indicate that only broad categories were enriched in the unique or shared genes between samples.”

# 4. Page 7. RNA sequencing. Which library kit was used to prepare the sequencing libraries? The authors state that mRNA libraries were generated so a rRNA depletion step was likely not required but how were those mRNA libraries generated? Different kits can have different biases and this is also important to know which adapters might be present in the sequencing datasets.

Libraries were prepared using a polyA capture library and a rRNA depletion step was not performed. We have added the following to the methods:

Lines 148-150

mRNA libraries were prepared by the UCSD IGM Genomics Center using Illumina Stranded mRNA Prep kit and sequenced on a single lane of an Illumina HiSeq 4000, using 100 bp paired-end reads.

# 5. Page 7. Microsporidia gene expression analysis. Were the adapters properly removed from the sequencing datasets prior to the analyses? The authors used two different methods to analyze RNAseq data, one for the microsporidia datasets and one for the nematode datasets (using the WormBase Alaska pipeline). The latter mention some sort of quality control (QC) but the first one does not mention any such QC step.

We used FastQC to perform the quality control on the microsporidia unaligned .fastq and Tophat-aligned .bam datasets. The adapters in the .fastq files were reported as overrepresented sequences by FastQC, but not in the .bam datasets, which indicates that they have been removed after alignment. We have added the following text in the methods section:

Line 157-159

“Using FastQC, quality control was performed on the Tophat aligned files to ensure that they are of good quality (Quality scores across all bases >30) and that the adapters were properly removed from the sequencing datasets prior to downstream analysis (Andrews, Simon, 2015).”

# 6. Page 7 lines 366-367: 'The paired end reads of each sample 10 hours post infection sample were submitted to Alaska v1.7.2 (http://alaska.caltech.edu)'.

What about the 20 and 28 PI samples? Were they also analyzed with the Alaska pipeline from WormBase? Did the authors mean: The paired end reads of each sample were submitted to Alaska v1.7.2 (http://alaska.caltech.edu)? The authors should also explicitly state what Alaska is. I had to reread that sentence twice as I found it confusing.

We have added the following text into the methods section to state what Alaska is, and explain why the 20 and 28 PI samples are not analysed with the Alaska pipeline:

Line 174-175:

“The paired end reads of each sample 10 hours post infection sample were submitted to Alaska v1.7.2 (http://alaska.caltech.edu), an online automatic C. elegans RNA-seq analysis pipeline.”

Line 181-182:

“The 20 and 28 hours samples were not submitted for analysis due to the lack of replicates at these timepoints.”

# 7. Page 7 methods and page 12 Fig 2C. From which RNAseq analysis method were the mapping metrics in Fig 2C derived; Alaska, the microsporidia one, both? A subtraction would not do here if contamination from a third party was present. Also, while I understand using the WormBase pipeline to analyze the Caenorhabditis data, this pipeline uses completely different tools than the ones used to analyze microsporidia data. Did the authors check for congruency between the two approaches?

We have added the following text to explain that the overall read mapping rate in Fig 2C and the description in page 7 methods are from the RNAseq analysis of microsporidia.

Line 154-157

“Reads from each sample were mapped to either Nematocida parisii strain ERTm1 (Genbank: GCF_000250985.1) or Nematocida ausubeli strain ERTm2 (Genbank: GCA_000250695.1) using TopHat v2.1.2, which also calculates the overall read mapping rate (Trapnell, Pachter and Salzberg, 2009).”

Figure legend of Figure 2C

“(C) Overall read mapping rate of Nematocida in each host at different time points, calculated by TopHat during alignment to the respective reference genomes.”

For the final point, Cufflinks had also been used previously for RNA-seq analysis of Nematocida (Cuomo et al., 2012). Since we did not directly compare the RNA-seq expression between the hosts and the microsporidian pathogens, we did not check for congruency between Alaska and Cufflinks. 

# 8. Page 8. Lines 420 - 424. 'From this analysis, we found additional domains and genes enriched with greater than three domain-containing proteins in any of our samples. These identified domains are chitinase-like (chil) proteins, CUB and CUB-like domains, cytochrome P450, glucosyltransferase family, nematode cuticle collagen N terminal domain, and UDP-glucuronosyltransferase'.

These sentences read like results rather than methods. Also, P450 cytochromes are involved in the detoxification of xenobiotics in insects. If those were enriched/overexpressed, something important might be going on here. Microsporidia spores are also made of chitin so chitinases appear highly relevant here.

To address the first point, we moved those sentences from the methods into the results:

Line 404-408

“From this analysis, we found additional domains and genes enriched with greater than three domain-containing proteins in any of our samples. These identified domains are chitinase-like (chil) proteins, CUB and CUB-like domains, cytochrome P450, glucosyltransferase family 92, nematode cuticle collagen N-terminal domain, and UDP-glucuronosyltransferase.”

We also added more text in the results elaborating the potential implication of enriched P450 cytochromes and chil genes:

Line 420-424

“Interestingly, cytochrome P450 genes, implicated to be involved in detoxification, are upregulated in both hosts, suggesting that the upregulated response to Nematocida overlaps with the metabolism and elimination of potentially toxic xenobiotics (Larigot et al., 2022). Moreover, we observe the upregulation of a few chil genes in C. elegans and C. briggsae, previously shown to be upregulated in response to epidermally infecting oomycetes (Osman et al., 2018; Grover et al., 2021). 

# 9. On page 1, lines 50-51. 'These two species of Nematocida are also commonly found to infect Caenorhabditis briggsae, which has been developed as a comparative species to C. elegans (Stein et al., 2003)'.

Weird phrasing: species do not arise to be compared to something and Caenorhabditis briggsae was not developed/created in lab (or was it?). I’m assuming that the authors meant that this species was used as an alternate model to facilitate comparative analyses or something like that…

We have modified the relevant text in the introduction:

Line 47-50:

“These two species of Nematocida are also commonly found to infect Caenorhabditis briggsae, a related species which has been used as a model organism to facilitate comparative analyses to C. elegans (Stein et al., 2003; Uyar et al., 2012).”

# 10. On page 2, lines 83-85 'A study looking at different species of bacterial infection in the nematode Pristionchus pacificus also showed both similarities and differences between the transcriptional responses in the two hosts (Sinha et al., 2012)'.

'showed both similarities and differences' This is vague. Minor/Major differences? A few but crucial ones?

To address this, we have described some major similarities and differences between the transcriptional responses of Pristionchus pacificus and C. elegans in the introduction:

Line 91-100

“A study looking at four different species of bacteria infecting the nematode Pristionchus pacificus also showed both similarities and differences between the transcriptional responses in the two hosts (Sinha et al., 2012). For example, while genes in the FOXO/DAF-16, TGF-beta and p38 MAP kinase pathways are suggested to be conserved between C. elegans and P. pacificus in response to bacterial infection, host-specific transcriptional responses are also observed, such as the enrichment of lipid metabolism related domains in the response of P. pacificus to different bacteria (Sinha et al., 2012).”

# 11. Figures 3 and 4 legends. Replace Microsporidia by Nematocida in the titles. These results are not representative of the whole group, and they would likely differ in non-Caenorhabditis hosts.

We have replaced “Microsporidia” with “Nematocida” in the legend titles of Figure 3 and 4.

# 12. Formatting: References are not formatted according to PLOS guidelines.

The references have now been formatted according to PLOS guidelines. 

Reviewer #2: The manuscript by Wan, Troemel and Reinke characterizes the microsporidian and host transcriptomes of two Nematocida spp. infecting two Caenorhabditis spp. The central questions posed by the authors are: 1. whether the parasites change their gene expression profiles depending on the host infected, and 2. whether the host gene expression profiles differ when infected by either Nematocida spp. The conclusion is that parasite gene expression profiles are very similar, no matter what host is being used, and that the same could be said for the host as well.

I think it is an interesting paper, well written and with very nice illustrations, which is worth of publication.

We thank the reviewer for their kind comments.

I have three questions to the authors that may (or may not) be explored in order to build an improved version of the manuscript.

1) Hosts were experimentally infected and RNA was extracted at 10, 20 and 28 hours post infection, with one replicate only for the 10 hpi timepoint. On line 257 of the discussion the authors recognize that the lack of replications could have limited their ability to detect differentially expressed genes. It would be interesting to know the extent by which their conclusions have been biased by the lack of replicates. Is it possible to estimate that in any way (e.g. by generating pseudo replicates)? Alternatively, would it be possible to provide at least one good argument to the reader giving support for their claims, i.e., that the conclusions are not simply a consequence of the lack of statistical power?

To address the possible bias by the lack of replicates, we provide the following arguments in the discussion:

Line 441-453

“Although a small number of replicates limits the ability to detect differentially regulated genes (Conesa et al., 2016), we believe that our conclusions are not simply a consequence of the lack of statistical power for the following reasons. Firstly, we saw a large overlap in significantly regulated genes compared to previously published transcriptional datasets of N. parisii infected C. elegans (Bakowski et al., 2014; Chen et al., 2017). The conclusions we reach in our study also agree with those from another transcriptional response comparison from our group that generated RNA-seq data in triplicate of C. elegans infected at a latter timepoint with four different species of microsporidia, including N. parisii and N. ausubeli (Mok et al., 2022). Secondly, previous statistical power calculations for RNA-seq analysis over a range of fold-changes and number of replicates suggests that a small number of replicates is sufficient to capture highly-expressed genes, such as the highly expressed IPR genes in our study (Conesa et al., 2016; Schurch et al., 2016; Lamarre et al., 2018). Finally, previously published transcriptional datasets from Bakowski et al., (2014) and Chen et al., (2017), which we compared our dataset to, were also performed in duplicates or triplicates.”

2) Another aspect that needs clarification is the statement that expression profiles are “evolutionarily conserved”. I am not sure whether you could say that for such closely related species, but the reader should at least know what evolutionary scale is actually being assessed in the study. Somewhere it should be mentioned to what degree both microsporidia and both hosts genetically diverge from each other (for any standard marker such as rRNA or other phylogenetic marker gene).

We have addressed these points above in the responses to reviewer #1. 

3) I am wondering whether it wouldn’t be possible, with the data at hand, to test other hypotheses. It would be interesting to know, for instance, what are the differences in expression 1. between parasites independently of the host, or 2. between hosts independently of the parasite. The first is going to be an unavoidable curiosity of the manuscript reader, considering that N. parisii and N. ausubeli differ in their infection phenotypes (lines 60-61).

As microsporidia only replicate inside of a host, it is not possible to measure the proliferative stage of infection independently. These parasites do exist outside of the host as dormant spores, but we did not measure mRNA from the spores. To clarify the life cycle of microsporidia infection more clearly, we have added the following text in the introduction:

Line 56-62

The infection cycle of N. parisii and N. ausubeli in Caenorhabditis hosts begins with ingestion of the microsporidia spores, an environmentally resistant and dormant stage of the pathogen (Troemel et al., 2008; Vávra and Ronny Larsson, 2014). In the intestine, the sporoplasm in the spore is ejected through the polar tube and deposited into the intestinal cells. This is followed by replication into meronts and differentiation into new spores, which then exit intestinal cells into the intestinal lumen. The spores are defecated by the nematodes, and the infection cycle repeats when they are ingested by another host (Troemel et al., 2008; Cuomo et al., 2012; Luallen et al., 2016). 

Although our data could be used to compare differences in expression between C. elegans and C. briggsae independent of infection, this was not a goal of our study. Additionally, a previous study (Grün et al., 2014) compared expression changes between C. elegans and C. briggsae at 7 time points. We have now added this reference to line 52.

---

## [Editor Report · Decision Letter 1]

1 Dec 2022

Conservation of Nematocida microsporidia gene expression and host response in Caenorhabditis nematodes

PONE-D-22-23486R1

Dear Dr. Reinke,

We’re pleased to inform you that your manuscript has been judged scientifically suitable for publication and will be formally accepted for publication once it meets all outstanding technical requirements.

Kind regards,

Nicolas Corradi

Academic Editor

PLOS ONE

Additional Editor Comments (optional):

Thank you for addressing all comments raised by the referees.
---

## [Editor Report · Acceptance letter]

6 Dec 2022

PONE-D-22-23486R1 

Conservation of *Nematocida* microsporidia gene expression and host response in *Caenorhabditis* nematodes 

Dear Dr. Reinke:

I'm pleased to inform you that your manuscript has been deemed suitable for publication in PLOS ONE. Congratulations! Your manuscript is now with our production department. 

Kind regards, 

on behalf of

Dr. Nicolas Corradi 

Academic Editor

PLOS ONE